# A metabolome atlas of the aging mouse brain

Jun Ding[1,2], Jian Ji[3], Zachary Rabow [1], Tong Shen[1], Jacob Folz[1], Christopher R. Brydges [1], Sili Fan[1], Xinchen Lu[1], Sajjan Mehta [1], Megan R. Showalter[1], Ying Zhang [1], Renee Araiza[4], Lynette R. Bower[4], K. C. Kent Lloyd [4] & Oliver Fiehn [1✉]

The mammalian brain relies on neurochemistry to fulfill its functions. Yet, the complexity of the brain metabolome and its changes during diseases or aging remain poorly understood. Here, we generate a metabolome atlas of the aging wildtype mouse brain from 10 anatomical regions spanning from adolescence to old age. We combine data from three assays and structurally annotate 1,547 metabolites. Almost all metabolites significantly differ between brain regions or age groups, but not by sex. A shift in sphingolipid patterns during aging related to myelin remodeling is accompanied by large changes in other metabolic pathways. Functionally related brain regions (brain stem, cerebrum and cerebellum) are also metabolically similar. In cerebrum, metabolic correlations markedly weaken between adolescence and adulthood, whereas at old age, cross-region correlation patterns reflect decreased brain segregation. We show that metabolic changes can be mapped to existing gene and protein brain atlases. The brain metabolome atlas is publicly available (https://mouse.atlas.metabolomics.us/) and serves as a foundation dataset for future metabolomic studies.

[1] West Coast Metabolomics Center, UC Davis Genome Center, University of California, Davis, 451 Health Sciences Drive, Davis, CA 95616, USA. [2] Department of Chemistry, Wuhan University, 430072 Wuhan, Hubei, P.R. China. [3] School of Food Science, State Key Laboratory of Food Science and Technology, National Engineering Research Center for Functional Foods, Synergetic Innovation Center of Food Safety and Nutrition, Jiangnan University, 214122 Wuxi, Jiangsu, P.R. China. [4] Mouse Biology Program, and Department of Surgery, School of Medicine, University of California, Davis, Davis, CA 95618, USA. ✉email: ofiehn@ucdavis.edu

The brain is one of the most structurally and functionally complex organs in mammals[1–4]. It is characterized by a unique morphology of anatomical regions, intricate brain network connectivity, an enormous variety of cell types, and a highly integrated molecular program during brain development and aging. Brain functions require a unique set of small molecule biochemicals, ranging from neurotransmitters to specific sets of complex lipids that are often enriched in highly unsaturated fatty acids (FAs). The brain has been mapped on the molecular level for gene transcript and protein expressions. In principle, metabolites represent the ultimate outcome of molecular biology. However, metabolite profiles cannot be simply predicted by genomic[5–10] or proteomic signatures[11] due to the multitude of feedback mechanisms and regulatory circuits. Surprisingly, the brain metabolome has been studied inadequately, either focusing on a few metabolites[12,13] or restricting study designs and data acquisition to incomplete coverage of neurochemistry and brain anatomical regions[14–17]. Today, the spatiotemporal metabolome of the mammalian brain remains incompletely understood.

We have started generating metabolome atlases of normal healthy populations to serve as a reference database for future studies. We here present the atlas of the aging mouse brain with an emphasis on the anatomical resolution of 10 brain regions and temporal coverage over the life period from adolescence (AD) to old age (OA). We identified 1,547 unique metabolites that cover many metabolic modules from nucleosides, lipids, and primary metabolites, such as glycolysis intermediates to neurochemicals such as acetylcholine, dopamine, and GABA. In addition, we created a freely accessible web tool to allow interactive exploration of the brain metabolome atlas at https://mouse.atlas.metabolomics.us/.

## Results

**Aging mouse brain metabolome atlas.** In collaboration with the UC Davis Mouse Biology Program, we have studied groups of 8 male and 8 female wildtype mice at AD (3 weeks), early adulthood (EA, 16 weeks), middle-age (MA, 59 weeks) and OA (92 weeks). Immediately after euthanasia, brains were harvested and then dissected into the following 10 anatomically defined regions: cerebral cortex (CT), olfactory bulb (OB), hippocampus (HC), hypothalamus (HT), basal ganglia (BG), thalamus (TL), midbrain (MB), pons (PO), medulla (MD), and cerebellum (CB). In total, these 640 brain samples were analyzed by combining data from three untargeted metabolomic platforms: primary metabolism by gas chromatography-time of flight mass spectrometry (GC-TOF MS) and two assays using orbital ion trap high-resolution mass spectrometry (Q-Exactive HF MS/MS) by separating biogenic amines using hydrophilic interaction chromatography (HILIC) and separating complex lipids by charged surface hybrid (CSH) reversed phase (RP) liquid chromatography (LC). Figure 1a illustrates this study design, with additional information given in Supplementary Data 1.

For obtaining a solid coverage of the brain metabolome, an untargeted LC–MS/MS assay used an iterative precursor mass exclusion[18]. LC–MS/MS raw data were processed by MS-DIAL[19], while GC-TOF MS data were processed by BinBase[20]. Using retention times and mass spectral information from the MassBank.us and NIST17 libraries, all mass spectra were manually investigated, yielding a total of 1,547 distinct annotated metabolites (Supplementary Data 2). In addition, concentrations for 853 metabolites were estimated based on internal standards, external calibrations and literature reports, summarized in Supplementary Data 3. This atlas represents the most comprehensive brain metabolome published so far, including MSI-compliant confidence levels[21,22]. We defined metabolites into eight chemical superclasses using the ClassyFire classification

system[23] (Fig. 1b). As expected, complex lipids accounted for the largest proportion of the brain metabolome due to the high endogenous contents of brain lipids ranging from phosphatidylcholines (PC), phosphatidylethanolamines (PEs), triacylglycerol (TGs), FAs, phosphatidylserines (PS), phosphatidylinositols (PIs), sphingomyelins (SM), ceramides (Cers) to diacylglycerols (DGs) and others. Organic acids including amino acids, modified amino acids, peptides and hydroxyl acids constitute 14% of the metabolome, while the remaining 15% was classified into organic oxygen compounds, organoheterocyclic compounds, benzenoids, organic nitrogen compounds, nucleosides, nucleotides and others. The vast majority of all brain metabolites were ubiquitously distributed across all ten brain regions to maintain essential brain functions (Fig. 1c). Complex lipids are hardly represented in standard biochemical pathway databases[24,25]. We therefore explored the pathway coverage of the annotated brain metabolome by querying HILIC and GC-resolved metabolites in Consensus PathDB (http://cpdb.molgen.mpg.de/)[26,27], comprising 118 pathway-based metabolite sets (Supplementary Data 4). The top-10 most important pathways are illustrated in Fig. 1d, indicating a sufficient breadth of pathway modules to interpret metabolic changes in brain regions during aging. Lipids were categorized into subclasses that reflect their chemical structure and function as given in the outer circle of Fig. 1b. The full dataset of the atlas can be downloaded as Supplementary Data 2.

**Quality assessment of the metabolome.** To assess the precision of the overall analytical method, a quality control reference pool sample (QC) was constructed from all brain extracts to reflect an aggregated brain metabolite composition. This pool QC was aliquoted and repeatedly injected between each set of 10 mouse brain samples. Data were utilized by the machine-learning-based SERRF software (https://slfan2013.github.io/SERRF-online/#)[28] to normalize metabolite intensities and correct for potential drift- or batch effects. Overall precision was then evaluated by analysis of the total variance using principal component analysis (PCA).

The PCA score plot in Fig. 2a showed that the QC samples were aggregated into a tight cluster near the origin of the plot, indicating minimal residual technical errors. Conversely, metabolic data of the different brain samples were scattered, explaining more than 43% of the total biological variance in the first two principal components. Using univariate analysis of all metabolites in the pool QC samples showed that nearly 60% of all annotated metabolites had excellent reproducibility with relative standard deviations (RSD) < 5%. A total of 91.3% of all metabolites were detected at RSD < 20%, highlighting a high quality of the data (Supplementary Fig. 1). Next, we investigated the reproducibility of metabolites of biological replicates of the different brain regions. We found strong Spearman-rank correlations within biological replicates of the same brain regions ($r_{xy}$ 0.46–0.90) and substantially lower correlations across different brain regions ($r_{xy}$ −0.63 to 0.63) (Fig. 2b). This finding indicated a good biological reproducibility of the dataset and distinct metabolic phenotypes of the different brain regions, captured by specific metabolite/metabolite correlation patterns[29,30].

**Regional metabolome architecture of the mouse brain.** When coloring the PCA sample plots by the different study parameters, i.e. brain regions, age, and sex, clear biological differences became apparent. Figure 2a showed that Principal Component 2 separated samples based on biological age with 12% of the total metabolic variance of the dataset, showing a clear difference between adolescent mice, early adult, and middle-aged mice, but also a significant difference between middle-aged and old mice. Importantly, we found only very minor differences in the brain metabolome between the sexes (Supplementary Fig. 2a)

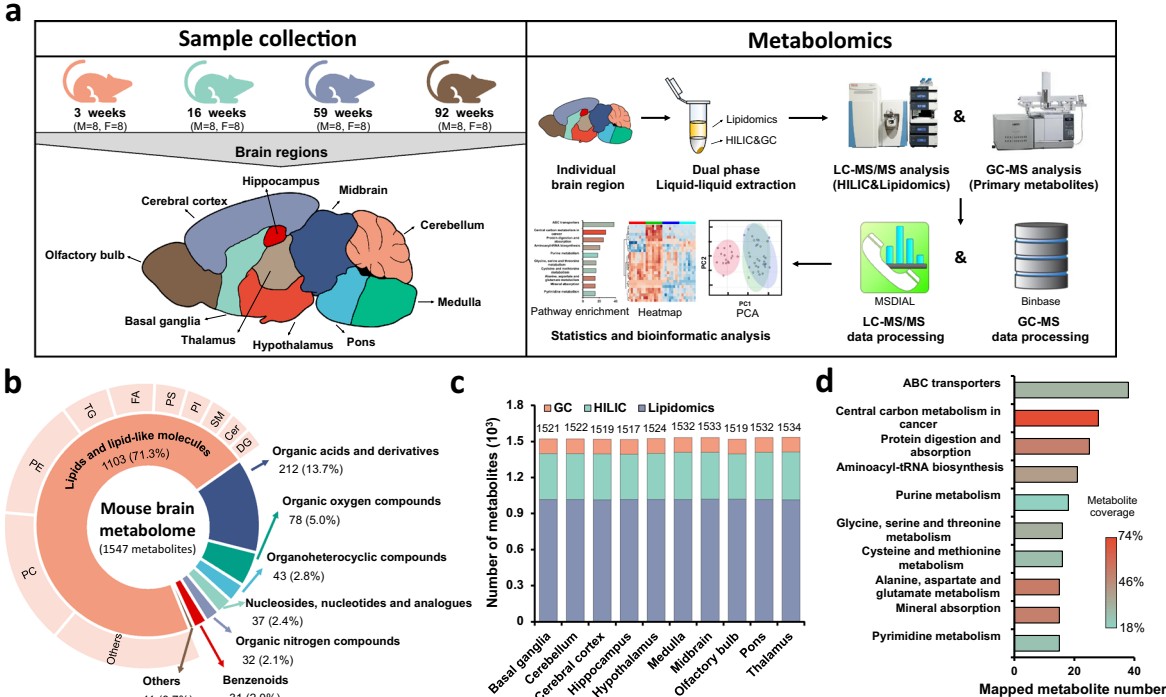

**Fig. 1 Overview of the mouse brain atlas dataset. a** Graphic illustration of the workflow to acquire aging mouse brain metabolome data. **b** Chemical composition of the mouse brain metabolome using ClassyFire categories to classify the metabolite diversity of all annotated metabolites across assays. **c** Number of annotated metabolites by metabolome assay and brain regions. **d** Mapping polar metabolites assayed by HILIC- and GC–MS to pathways. Top-10 mapped pathway-based sets shown from a total of 118 pathways covered by Consensus PathDB.

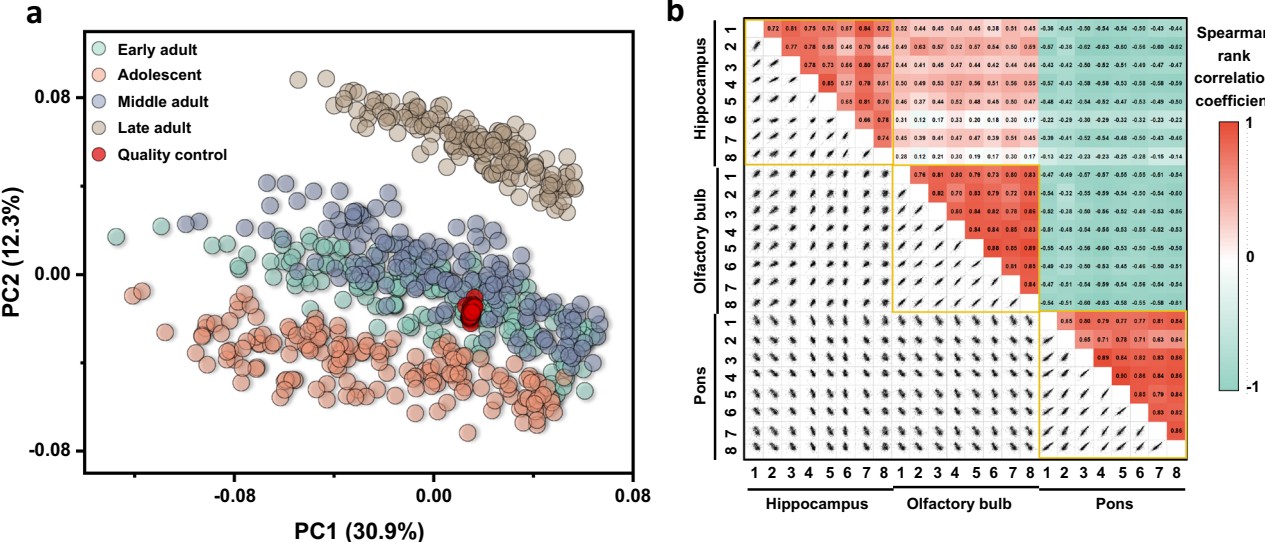

**Fig. 2 Data quality assessment of the mouse brain metabolome. a** Principal component analysis (PCA) of 640 mouse brain metabolome samples and corresponding Quality Control samples (QC, labeled red). QC samples are highly clustered, demonstrating low technical variance and high reliability of metabolomic analyses. **b** Quality control analysis by Spearman rank analysis testing the hypothesis that metabolic correlations within brain regions should be larger than correlations across brain regions. 276 scatter plots from biological replicates of three brain regions confirm this biological quality trait. Each scatter plot correlates peak intensities of all annotated metabolites in one sample against those in another sample. Output Spearman rank correlation coefficients are given in the top-right heat map.

suggesting that brain chemistry is mostly spatially and temporally regulated but not by sex hormones. Figure 3a was colored indicating the metabolic phenotypes of different brain regions, with PC1 explaining 31% of the total variance, separating the CT (gray) versus brainstem tissues (blue colors) with the HC (red) and MB tissues (dark blue) in between. A clearer distinction of

metabolic phenotypes was obtained when only tissues from the early adult group were analyzed (Fig. 3b, other age groups see Supplementary Fig. 2b–d). Here, diverging sample clusters were obtained with cerebrum tissues mapped adjacent to each other (CT, OB, HC, HT, BG) and clearly separated from brainstem tissues (MB, PO, and MD) by Principal Component 1. CB was

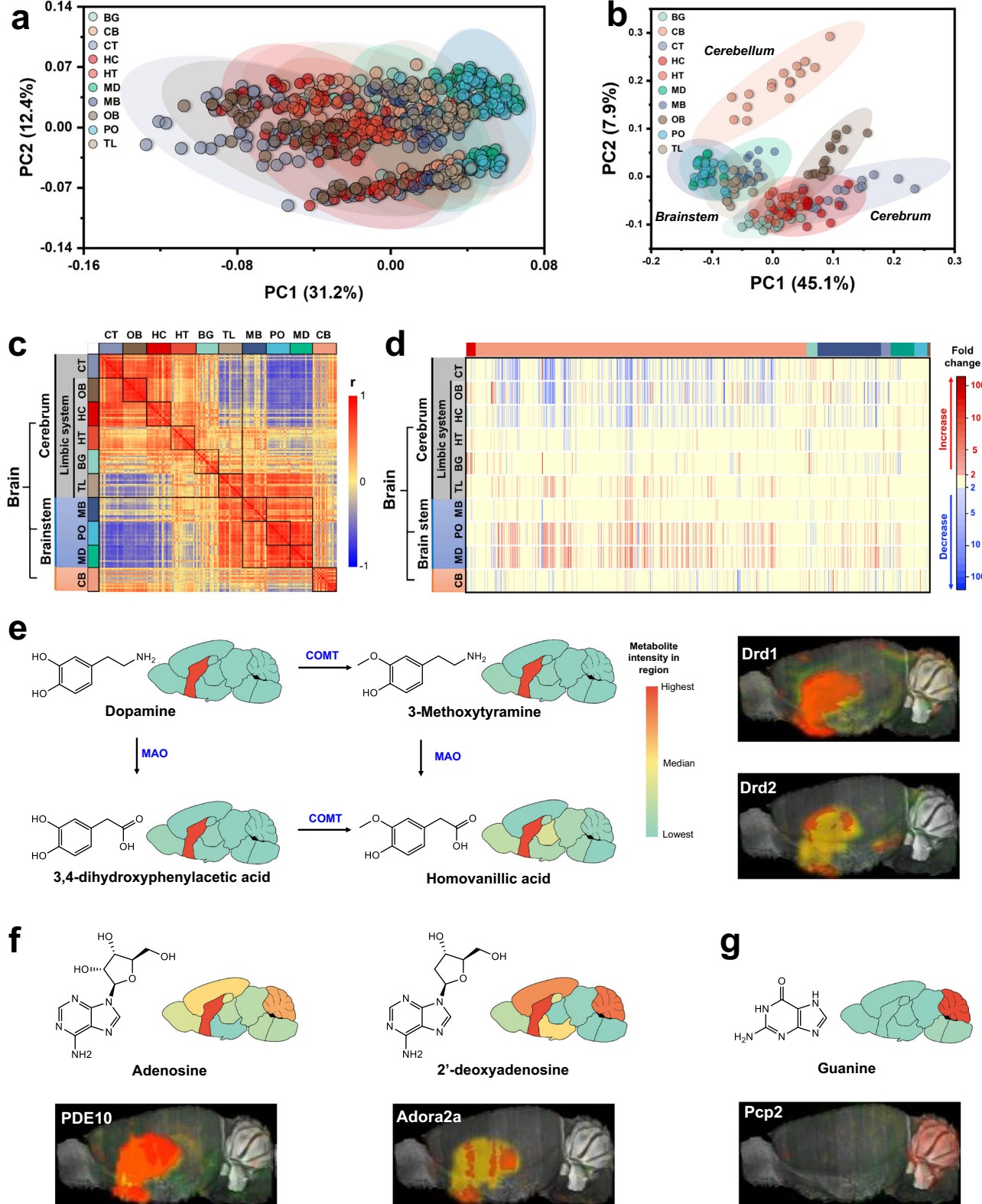

separated by PC2, discovering a similar biochemical distinction as observed in a previous brain proteome study[11]. Hence, metabolic phenotypes follow the major structural divisions of classic brain anatomy. TL samples were located between cerebrum and brain stem clusters, reflecting the TL function to be highly interconnected with both cerebrum and brainstem tissues to relay incoming information to nearby brain regions[31]. Similar

molecular differentiation of brain regions has been observed in brain genomic maps[7,9,10], suggesting a consistency of genomic, transcriptomic, and metabolomic patterns in brain regionalization and development.

Next, we exploited the correlative nature of brain metabolites to investigate specific metabolic differences across brain regions. A detailed analysis of the Spearman-rank correlation matrix

**Fig. 3 Regional biochemical differences of the mouse brain. a** Principal component analysis (PCA) of all mouse brain metabolome samples. PCA vector 1 separates samples into different brain regions. Samples are colored by brain regions. **b** Principal component analysis (PCA) focused on early adult mice for all 10 brain regions. PCA vector 1 scparates cerebrum and brainstem, vector 2 distinguishes the cerebellum from cerebrum and brainstem samples. Samples are colored by brain regions. **c** Heatmap matrix of pairwise Spearman correlations between brain regions in early adults. Strong correlations are given in red, strong negative correlations in blue. Overall correlation structures distinguish the three main brain divisions cerebrum, brainstem, and cerebellum. **d** Heatmap of metabolites differentially expressed across the different brain regions, constrained to metabolites with >2-fold changes. Metabolites are categorized by ClassyFire. From left to right: Benzenoids: red, Lipids: orange, Nucleosides: light green, Acids: dark blue, Nitrogen organics: purple, Oxygen organics: dark green, Heterocyclics: light blue, Others: dark gray. **e** Co-localization maps of dopamine metabolites and in situ hybridization of dopamine receptors. Drd1 and Drd2 in situ hybridization images are taken from the 2004 Allen Institute for Brain Science (http://mouse.brain-map.org). Image credit: Allen Institute. **f** Co-localization maps of adenosine metabolites and in situ hybridization of adenosine receptors Adora2a and cAMP hydrolase PED10. The Adora2a and PED10 in situ hybridization images are taken from the 2004 Allen Institute for Brain Science (http://mouse.brain-map.org). Image credit: Allen Institute. **g** Co-localization maps of guanine and in situ hybridization of the guanine nucleotide dissociation inhibitor Pcp2. The Pcp2 in situ hybridization image is taken from the 2004 Allen Institute for Brain Science (http://mouse.brain-map.org). Image credit: Allen Institute.

(Fig. 3c) across all 10 brain regions revealed highly positive metabolic correlations of the metabolome within each brain architectural division, but negative correlations across the divisions of cerebrum, brainstem, and CB tissues[5]. Specifically, the negative correlations between the cerebrum and brainstem metabolome mirrored regionally distinct functions, possibly because the cerebrum have the most complex neuronal network to direct higher cognitive functions, whereas the evolutionary oldest brainstem regulates autonomous activities. Hence, the major brain divisions were functionally separated by metabolome phenotypes. An overview of regionally differentiated metabolites at minimum 2-fold changes and FDR $p < 0.05$ is displayed in the heatmap Fig. 3d. Metabolite changes were found distributed across all brain regions and were not dominated by any single specific region. Interestingly, distinct metabolic heterogeneities became visible. Lipid profiles in the brainstem PO and MD were distinctly upregulated in comparison to all other regions, while in the cerebrum, CT and OB displayed many lipids at overall decreased levels. We associated brain regions with specific metabolites using 5-fold differences in abundance and FDR significance $p < 0.05$ compared to the average of all other regions (Supplementary Data 5). For example, PO and MD were uniquely enriched in SM and Cers, while the OB showed cholesterol esters to be manifold-more abundant than other regions. BG were uniquely enriched in aromatics.

We provide an interactive web tool https://mouse.atlas.metabolomics.us/ to investigate specific metabolites in the mouse brain atlas. These metabolome maps enable users to visualize levels of metabolites-of-interest across 10 anatomical regions, four life periods, and both sexes. In this way, users can generate or verify hypotheses regarding brain metabolism, for example, by comparison to other imaging resources. In Supplementary Fig. 2e this possibility is shown for the neurotransmitter acetylcholine that exerts clear regional enrichment in BG. This regiospecificity can be readily associated with the endogenous brain enzymes responsible for acetylcholine synthesis and degradation. Acetylcholine is formed by esterification of acetic acid and choline catalyzed by Choline Acetyltransferase (Chat) which is highly expressed in BG[2]. Acetylcholine is then transported via vesicular transport of cholinergic neurons to CT, MB, and HC (Supplementary Fig. 2e) as well as to smaller regions like the BG, TL, and HT. Acetylcholine transport is further assisted by Chat expression in BG, MB, and TL, validating our metabolic brain map by in situ hybridization maps[4]. Cholinergic neurons in the basal forebrain are densest in the cortex area[32], which is supported by the medium acetylcholine levels throughout this brain region. Acetylcholine is degraded by Acetylcholinesterase (AChe) enzymes and has a half-life of 1–2 ms in the brain[33]. In situ hybridization of AChe shows its enrichment in MD, PO, MB, and depletion of the enzyme in OB and CB[4]. Accordingly, our brain map shows the lowest acetylcholine

abundances in the brainstem regions (MD and PO), and CB has the lowest levels of acetylcholine in the brain. This finding is validated by the low density of cholinergic markers in the CB in multiple animal species, including neurons, AChe, and Chat enzymes[34]. Similarly, the brain metabolome atlas shows high enrichment of the neurotransmitter dopamine and its metabolites in BG. This finding is consistent with in situ hybridization results for the dopamine receptors Drd1 and Drd2 in the Allen Brain Atlas project (http://mouse.brain-map.org)[5] (Fig. 3e). Likewise, adenosine and its analog are highly abundant in BG and in full accordance to *in situ* hybridization images of the adenosine A2a Receptor (Adora2a) and phosphodiesterase 10 (PED10, a cAMP hydrolase)[5] (Fig. 3f). Crucially, matching highly abundant metabolites and gene expression can contribute to the verification of gene functions. We here show that guanine is highly expressed in the CB and is exactly co-localized with the Purkinje cell protein 2 (Pcp2) (Fig. 3g)[5]. This co-localization in two different mouse brain maps gives extraordinary evidence for the proposed function of Pcp2 as guanine nucleotide dissociation inhibitor[5,35,36]. Apart from in situ hybridization, other brain maps can be used such as functional magnetic resonance imaging (fMRI), genomics[5–7], transcriptomics[8–10], proteomics[11] or synaptomics[4], offering a practical resource to integrate brain metabolism into systems biology.

**Aging impact on the mouse brain metabolome.** During aging, the brain undergoes a series of changes in structure and function. The underlying molecular program has recently been detailed by genome-[37] and transcriptome-[9,10] based maps. However, it remains incompletely understood how aging impacts the brain metabolome. The PCA plot in Fig. 4b indicated substantial differences in metabolome architecture between different ages. To visualize the corresponding difference in metabolic regulations, we plotted the correlation matrices of brain regions at each age (Fig. 4a). In the transition from adolescent to early adult mice, a large shift from highly positive to highly negative correlations is observed for brainstem versus cerebrum. Simultaneously, internal correlations within the different cerebrum regions (TL BG, HT, HC, OB, CT) weaken significantly. Interestingly, similar patterns have been reported in fMRI studies in the cortical-subcortical (limbic) and between-subcortical functional connections[38–40]. This differentiation was interpreted as an intrinsic feature of the maturation of the adolescent to allow for functional specialization, reducing interregional interference and facilitating cognitive performance in the coming adulthood[41]. The age-dependent metabolome dynamics provide molecular support for this interpretation. In the aging process from EA to middle age, even the strong negative correlations between brainstem and cerebrum regions are severely diminished. At OA, almost all negative correlations have disappeared (Fig. 4a), possibly due to increased lateral diffusion and enlargement of brain ventricles, and lesser

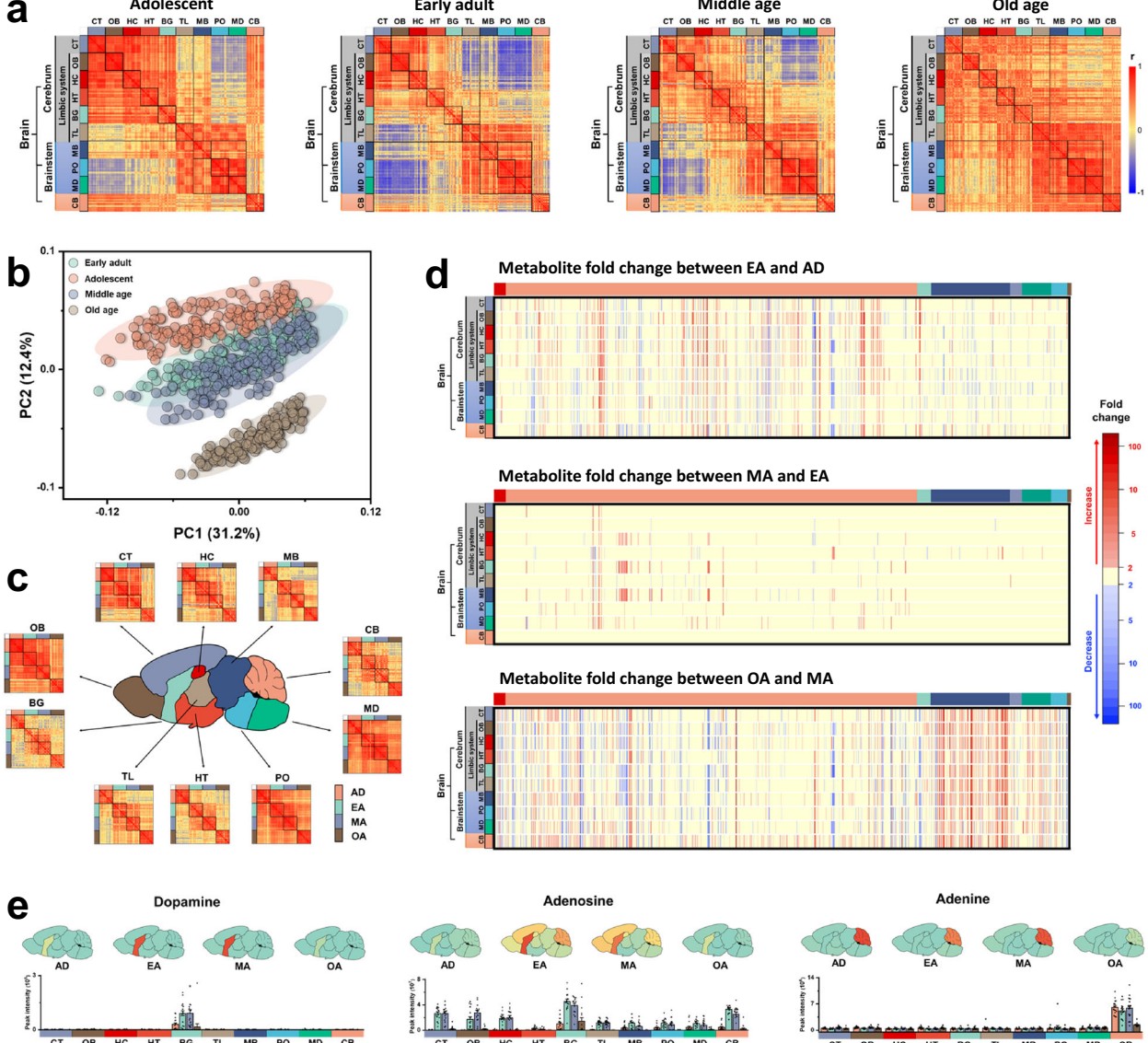

**Fig. 4 Impact of aging on the mouse brain metabolome. a** Correlation matrices for brain regions of 16 mice from adolescent (AD), early adult (EA), middle-age (MA) to old age (OA) groups across brain divisions (brainstem, cerebrum, cerebellum) and 10 brain regions. Positive correlations, red, negative correlations, blue. **b** Multivariate analysis of mouse brain metabolomes by principal component analysis. PCA vector 2 separates samples into different ages. Samples are colored by age groups. **c** Correlation heatmaps for individual brain regions reveal different developmental patterns during aging. **d** Heatmap of metabolites with >2-fold changes between age groups. Metabolites are categorized by ClassyFire. From left to right: Benzenoids: red, Lipids: orange, Nucleosides: light green, Acids: dark blue, Nitrogen organics: purple, Oxygen organics: dark green, Heterocyclics: light blue, Others: dark gray. **e** Visualization of brain maps and bar plots across age groups when querying three selected metabolites at mouse.atlas.metabolomics.us. $n = 16$ biologically independent samples. Data are presented as mean values ± SEM.

rejuvenation of neuronal cells in rodents[42]. This decrease in regional segregation and modularity of the brain may be associated with a decline in cognitive functions at very OA[43]. Combined, these processes suggest an increase in specificity of brain functions during the maturation age, and a de-differentiation during aging. Similar trajectories have been observed in the brain transcriptome map[10], the synapse atlas[4], and brain functional connectivity[43].

Previous studies showed that brain development and aging occur asynchronously in a region-specific manner instead of uniformly throughout all regions[42,43]. We find similar aging trends in the metabolome correlation heatmaps for each region (Fig. 4c). For example, the correlation matrices for OB, PO, and MD metabolomes remain robust across all studied ages.

Conversely, metabolic regulations are drastically changing in the transition from adolescent to early adults in CT, HC, HT, BG, TL, MB, and CB regions, remain stable during adulthood, and keep changing at OA. These findings verify distinct programs of brain development on the metabolome level. A possible reason could be that metabolic needs of OB, PO, and MD regions are required to be kept at steady levels to execute normal neurophysiological activities for basic and vital functions such as respiration, movement, smell, and senses[44]. Other regions are responsible for higher cognitive and social abilities, such as executive, emotional, and decision-making functions. Changes in the cerebral and CB metabolome are required to cope with these substantial changes in life during brain development and aging[45].

These differences in metabolic regulations are also visible by changes in individual metabolite levels. In the transition from adolescent to early adults, 37.1% of all metabolite levels significantly change, only 10.0% of all metabolites show such changes between EA and middle age. Strikingly, 61.5% of all metabolites are altered from middle age to OA (Fig. 4d). The shift patterns of metabolites during aging were illustrated by both metabolite classes and unsupervised hierarchical clustering (Fig. 4d and Supplementary Fig. 3). Complex lipids and nucleosides metabolism are continuously affected at all ages, possibly caused by the ongoing synaptic pruning and myelination remodeling throughout life[4,46]. At OA, very pronounced effects were found on neutral lipids. A higher degradation of TGs to DGs was found concomitant with an increase in monounsaturated FAs, likely due to increased activity of lipases. On the other hand, acylcarnitines increased along with a decrease in free saturated FAs, possibly due to lower mitochondrial oxidation at OA. Similarly, mitochondrial alpha-ketoglutarate dehydrogenase showed lower activity causing decreased succinate levels with increased alpha-ketoglutarate levels. This lower activity in mitochondria at OA may also be reflected by the observed increases of sugars, sugar phosphate, and pyruvate as the end product of glycolysis. This metabolic shift at OA was not due to differences in vitamin levels such as pantothenate or thiamine. Interestingly, no changes in epimetabolites were observed either, such as methylated-, acetylated- or oxidized metabolites. Structural degradation of brain matter was more pronounced at OA with increased protein breakdown associated with increased levels in amino acids and di- and tripeptides. On the contrary, old mice showed lower levels of neurotransmitters such as acetylcholine and dopamine, along with metabolites with neuronal signaling functions such as adenosine and indoxyl sulfate. All these metabolic changes may result in cognitive decline and increased vulnerability to neurodegenerative diseases. A considerable number of metabolites are co-upregulated or downregulated across different brain regions in a synchronous manner, consistent with brain transcriptional dynamics over aging[10]. For example, phosphatidylmethanols first decrease, then increase, and decrease again over aging, and these changes are synchronous across all 10 brain regions. Short-chain PIs (C28–C34) persistently decrease in almost all regions from AD to middle age and then increase at OA. Conversely, adenosine and its related metabolites increased from AD to EA in most regions but kept declining afterward. Such aging-induced patterns of many metabolite classes have not been reported before. The aging profile for each metabolite is readily visualized in our web tool, with corresponding bar graphs giving both peak intensities and statistical significances between age groups and brain regions. For example, we herein show the aging difference of dopamine, adenosine, and guanine across brain regions in Fig. 4e.

**Sphingolipid dynamics indicates myelin remodeling in the adult brain**. Next, we studied if the mouse brain metabolome atlas yields insights into biological processes. We found an interesting age-dependent alternation pattern of three sphingolipid species, including SMs, hexosylceramides (HexCers), and their sulfatides (sHexCers). Heatmaps of SMs, HexCers, and sHexCers show large increases from adolescent to early adults and substantial decreases from middle age to OA (Fig. 5a), specifically for HexCers and sHexCers with C20–26 fatty acyl groups. This finding is consistent with the up-regulation of ceramide synthase 2 (CerS2) (Fig. 5b) in 3-week-old mouse brains whereas a down-regulation in 60-week-old brain[47,48], an enzyme

predominantly expressed in oligodendrocytes to produce C20–26 Cers. Several HexCers continued to increase in metabolite levels from early adult to middle age and decrease afterward. These characteristic changes were structurally confined to C22–24 monounsaturated fatty acyl groups. For sHexCers, the highest levels were maintained at the middle-age stage, specifically for odd-chain C23–C25 monounsaturated fatty acyl groups. Afterward, several components, especially the C24 saturated species, began to decrease. Interestingly, this finding is supported by a CerS2 knock-out study that revealed unstable and non-compact myelin in late adult knock-out mice[49]. Similarly, both aging brains and brains with Alzheimer's disease undergo loss of CerS2 activity accompanied by myelin degeneration[48], confirming the importance of very long acyl chain species for the maintenance of myelin function and integrity in the brain. Only four SMs showed significant large fold changes between middle age and OA with large differences between regions, during aging, and between the sexes via web-based visualization. Other sphingolipids showed decreases during aging, for example, short-chain fatty acyl derivatives, specifically for HexCer and sHexCer. A consistent increase of all HexCer species was witnessed only in CT at OA. SM and sHexCer were most abundant in brainstem regions (MB, PO, and MD) and showed fewer drastically altered metabolites between adults and OAs. This result is in accordance with a previous study that showed characteristic spatiotemporal myelination patterns[50].

HexCers, sHexCers, and SMs are highly enriched in oligodendrocytes or myelin[51]. Myelins are an indispensable structure in the central nervous system to insulate neuronal axons for the acceleration of neuronal transmission and maintenance of neuronal function. HexCers are formed via glycosylation of Cers and can be further sulfated (Fig. 5b), while SMs are generated via hydrolysis of Cers or by synthesis using phosphatidycholine. The structural diversity of these sphingolipids is regulated by the expression of different ceramide synthases via the incorporation of FAs with different acyl chain lengths. The concentration of HexCers, sHexCers, and SMs in the brain is proportional to the amount of myelin present during brain development[52]. Short-chain and long-chain sHexCers have been found to be representative of different developmental stages of oligodendrocytes[53]. Brains of newborns are virtually unmyelinated, and the oligodendrocyte population expands dramatically to form myelin after birth to the first few years of childhood[54], while less is known about myelination after brain maturation or aging. Recent fluorescence optical studies suggest that myelin may be remodeled to support the circuit plasticity of the brain after AD[55–57], specifically in the cortex and subcortex. In aged brains, abnormality and loss of myelin have been identified to result in cognitive decline and a higher risk of neurodegenerative diseases[58]. The mouse brain metabolome atlas adds molecular details to such overall processes.

We here show how the myelinating process is evolving from AD to OA. Very specific acyl-chain lengths are specifically affected for HexCer, sHexCer, and SM, with distinct patterns for acyl-groups longer than 18 carbons (and long-chain SMs) (Fig. 5c), usually in a monounsaturated form. Such structures may support myelin plasticity in the adult or aging brain[57]. Cell–cell adhesion in myelin is partially due to sphingolipid carbohydrate–carbohydrate interactions across opposed membranes[59]. We hypothesize that during the transition from AD to adulthood, shifts in levels of very-long acyl chain sphingolipids lead to increases in sphingolipid hydrophobic properties to further strengthen the stability of myelin structures. In contrast, at OA the decrease of very-long acyl chains in sphingolipids lowers the hydrophobicity of myelin, resulting in myelin degeneration and cognitive decline at OA. Similar shifts were reported for 12 sHexCers by matrix-assisted laser Ionization mass spectrometry that focused on cell-cultured oligodendrocyte development[53].

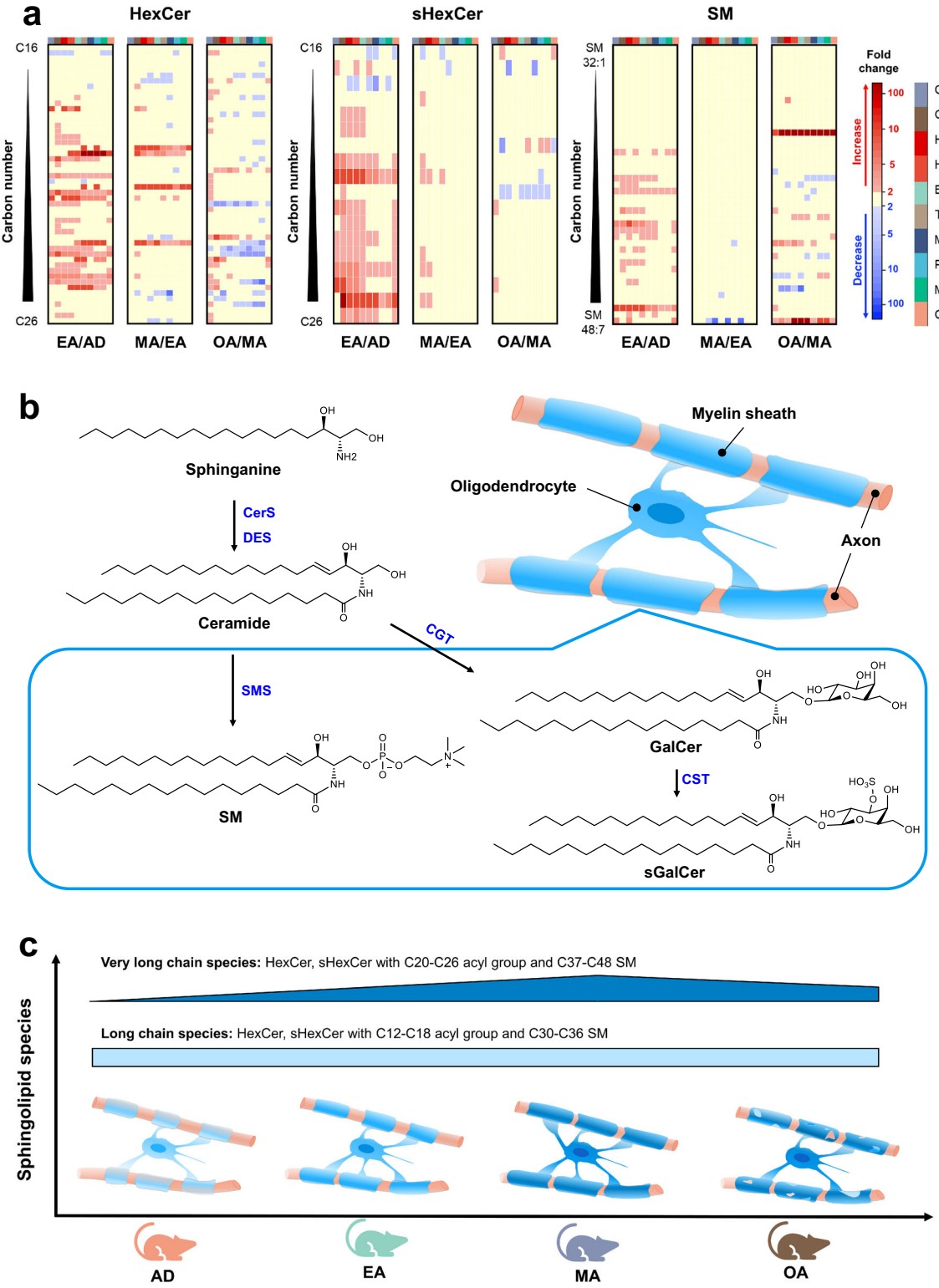

**Fig. 5 Dynamics of sphingolipids in the aging mouse brain. a** Heatmaps with fold-changes for HexCer, sHexCer, and SM sphingolipids in brain regions between early adult versus adolescent, middle-age versus early adult, and old age versus middle age. **b** Pathways of HexCer, sHexCer, and SM biosynthesis. HexCer, sHexCer, and SM are highly enriched in oligodendrocyte or myelin. **c** Simplified scheme summarizing myelin sphingolipid changes during brain aging. Very-long-chain sphingolipids increasing from adolescent to middle-aged brains with a subsequent decrease from towards old age. Long-chain sphingolipids almost keep constant levels across all age groups.

## Discussion

Understanding brain function must start with an in-depth mapping of the brain's structural and molecular organization. With 1,547 annotated metabolites across 10 brain regions, we here present a large-scale comprehensive metabolome atlas of the aging mouse brain that can inform previously established genomic, transcriptomic and proteomic atlases[5–11]. Data analysis revealed that metabolic differences were almost entirely found between brain regions and along the axis of brain maturation and aging, but not due to sex differences. Hence, these metabolomic differences were in great concordance with functional and molecular phenotypes published before[1,2] that highlighted large differences during brain aging and between brain regions, validated also by concordance between metabolite abundances and enzyme imaging techniques[4]. This atlas can therefore serve as an important repository for the health status of different mouse brain regions against which diseased states or the impact of mutations can be compared in future studies, and for research use into dementia and other brain dysfunctions associated with very OA.

The differentiation–dedifferentiation trajectory of brain development indicates specific aging programs for each region[4,43]. We found this trajectory to be reflected by decreasing metabolite correlations across brain regions when comparing adolescent to aged mouse brains. We exemplified the use of the brain metabolome atlas by highlighting novel metabolic patterns of HexCers, sHexCers, and SMs during brain development and aging. Here, we contribute previously unknown details on the exact molecular speciation of sphingolipids localized in oligodendrocytes and myelin remodeling during adulthood and OA[47,48].

In general, we did not find a large variance in metabolic levels for the eight samples per sex and brain region for each age group. In addition, few differences in brain metabolism were observed between the sexes. Hence, overall the study appeared to be not biased by the number of samples. Yet, this brain metabolome atlas will be further extended in the future. For example, changes during the prenatal and postnatal periods might be highly interesting but would require even more advanced techniques for microdissection and metabolome analysis. Changes in blood and fecal metabolomes during the aging process may also provide an auxiliary impact on changes in brain function. Furthermore, the spatial organization of brain functions clearly continues from brain regions to the cellular and subcellular levels. Here, other techniques such as laser-based mass spectrometry imaging are useful[60], especially once these methods extend from lipids to polar metabolites. However, what is gained by such techniques in spatial resolution is often lost with respect to molecular speciation and statistical power for finding subtle differences. Our methods also yielded several thousands of structurally yet unidentified compounds. We anticipate that extended mass spectral libraries[61,62] and improved retention time predictions[63,64] as well as novel separation techniques such as ion mobility[65] will increase the number of identified compounds in the future. We will then update the brain metabolome atlas accordingly. Last, all biochemical findings are based on mouse brains. Underlying molecular mechanisms remain inadequately understood, and we should be very careful when extrapolating findings from rodent animal models to humans.

## Methods

**Tissue collection**. Mice were cohoused by gender groups of 4–5 in individually ventilated cages (Optimice IVC, Animal Care Systems, Centennial, CO) on a 12:12-h (6:00/18:00) light:dark cycle at 68–79 °F with 40–60% humidity and provided water and standard rodent chow (Rodent chow, Harlan 2918) ad libitum. Brain tissue samples were collected from 3, 16, 59, and 92 weeks old male and female wild-type mice on a C57BL/6N background. All procedures were approved by the IACUC of the University of California, Davis, which is an AAALAC-accredited institution. Animal housing and euthanasia were performed in accordance with the recommendations of the *Guide for the Care and Use of Laboratory Animals*. Briefly, mice

were anesthetized with 4% Isoflurane in 100% oxygen at a flow rate of 3 L/h to a surgical plane. Blood was then collected by a retro-orbital bleed into an EDTA tube and centrifuged at 2000 × *g* for 15 min to separate and remove plasma. While under anesthesia mice were perfused for ~10 min with phosphate-buffered saline (PBS) pH 7.4 at room temperature. Following perfusion, the brain was removed and placed in a Petri dish containing PBS at 4 °C for dissection of individual brain regions. A dissection microscope, fine tip (#5) forceps, and razor blade were used to isolate and separate brain regions (OB, HC, HT, TL, MB, CB, PO, MD, CT, and BG collected as caudate-putamen and basal forebrain) in induvial mice while being careful to avoid contamination from neighboring regions. Briefly, after separating the OBs, the left and right cerebral cortices were then removed while taking care not to disrupt the regions underneath. This enabled access to and removal of the left and right HC. After cutting along the TL, the left and right caudate putamen was separated and removed from the basal forebrain. Subsequently, the CB and MB were isolated and removed, followed by separation and removal of the TL and the HT from the PO and MD. The PO was then separated from the MD. Any spinal cord remaining on the MD was removed. Each region was immediately placed in a cryovial and flash-frozen liquid nitrogen for analysis.

A full necropsy, including brain dissection and isolation and removal of tissues, took ~25 min on each mouse. Retro-orbital bleed and perfusion were completed in the first 15 min, and the brain dissection was finished in the remaining 10 min.

**Sample preparation**. During the sample preparation, lipids and polar metabolites were separated prior to analyses through solvent extraction/fractionation. Hence, potential problems in ion suppression or the ability of compounds to be ionized were limited due to the use of Lipidomic LC–MS/MS, HILIC-MS/MS (including positive and negative electrospray), and the complementary use of GC-electron ionization-MS.

Briefly, five milligrams of tissue from each brain region were homogenized in 225 μL of −20 °C cold, internal standard-containing methanol using a GenoGrinder 2010 (SPEX SamplePrep) for 2 min at 1,350 rpm. The extraction methanol contained the following internal standards for quality control and retention time normalization: sphingosine (d17:1), LPE (17:1), LPC (17:0), MG (17:0/0:0/0:0), DG (12:0/12:0/0:0), PC (12:0/13:0), cholesterol-$d_7$, SM (18:1/17:1), ceramide (d18:1/17:0), PE (17:0/17:0), TG (14:0/16:1/14:0)-$d_5$, TG (17:0/17:1/17:0)-$d_5$, acylcarnitine (18:1)-$d_3$, fatty acid (16:0)-$d_3$, MAG (17:0/0:0/0:0), PI (15:0–18:1)-$d_7$, PG (17:0/17:0), PS (15:0-18:1)-$d_7$, glucosylceramide(d18:1/17:0), mono-sulfo galactosylceramide(d18:1/17:0), and 5-PAHSA-$d_9$. The homogenate was vortexed for 10 s. 750 μL of −20 °C cold, internal standard-containing methyl tertiary-butyl ether (MTBE) was added, and the mixture was vortexed for 10 s and shaken at 4 °C for 5 min with an Orbital Mixing Chilling/Heating Plate (Torrey Pines Scientific Instruments). MTBE contained cholesteryl ester 22:1 as internal standard. Next, 188 μL room temperature water was added and vortexed for 20 s to induce phase separation. After centrifugation for 2 min at 14,000× *g*, two 350 μL aliquots of the upper non-polar phase and two 125 μL aliquots of the bottom polar phase were collected and dried down. The remaining fractions were combined to form QC pools and were injected after every set of 10 biological samples.

The non-polar phase employed for lipidomics was resuspended in a mixture of methanol/toluene (60 μL, 9:1, v/v) containing an internal standard [12-[(cyclohexylamine) carbonyl]amino]-dodecanoic acid (CUDA)] before injection. Resuspension of dried polar phases for HILIC analysis was performed in a mixture of acetonitrile/water (90 μL, 4:1, v/v) containing the following internal standards: CUDA, caffeine-$d_9$, acetylcholine-$d_4$, TMAO-$d_9$, 1-methylnicotinamide-$d_3$, Val-Tyr-Val, betaine-$d_9$, acyl carnitine (2:0)-$d_3$, N-methyl-histamine-$d_3$, L-carnitine-$d_3$, butyrobetaine-$d_9$, L-glutamine-$d_5$, aspartic acid-$d_3$, L-arginine-$^{15}N_2$, cystine-$d_4$, asparagine-$d_3$, histidine-$d_5$, isoleucine-$d_{10}$, leucine-$d_{10}$, methionine-$d_8$, ornithine-$d_2$, phenylalanine-$d_8$, proline-$d_7$, threonine-$d_5$, tryptohan-$d_8$, tyrosine-$d_7$, valine-$d_8$, spermine-$d_8$, glucose-$d_7$, fructose-6-phosphate-$^{13}C_6$, succinic acid-$d_4$, taurocholic acid-$d_4$, adenosine 5′-monophosphate-$^{15}N_5$, uridine 5′-monophosphate-$^{15}N_2$, dopamine-$d_4$, taurine-$d_4$, uracil-$d_2$, biotin-$d_4$, N-acetylalanine-$d_3$, guanine-$^{13}C$, and adenosine-$^{13}C_5$. The second dried polar phase was reserved for GC analysis and a following derivatization process was carried out before injection. First, carbonyl groups were protected by methoximation with methoxyamine hydrochloride in pyridine (40 mg/mL, 10 μL) was added to the dried samples. Then, the mixture was incubated at 30 °C for 90 min followed by trimethylsilylation with N-methyl-N-(trimethylsilyl) trifluoroacetamide (MSTFA, 90 μL) containing C8–C30 fatty acid methyl esters (FAMEs) as internal standards by shaking at 37 °C for 30 min.

**Lipidomic LC–MS/MS analysis**. For lipidomics analysis, 3 μL of the resuspended non-polar phase was injected into a Vanquish UHPLC system (Thermo Scientific, Waltham, MA, USA) equipped with a Waters Acquity UPLC CSH C18 (100 mm × 2.1 mm i.d.; 1.7 μm) coupled with a Waters Acquity VanGuard CSH C18 precolumn (5 mm × 2.1 mm i.d.; 1.7 μm). The oven temperature and flow rate were set at 65 °C and 0.6 mL/min, respectively. In order to obtain a broad lipid coverage, different mobile phases were employed for positive mode and negative mode analysis, respectively. The positive mobile phase consists of acetonitrile/water (60/40, v/v) with 0.1% formic acid and 10 mM ammonium formate as A and 2-propanol/acetonitrile (90:10, v/v) with 0.1% formic acid and 10 mM ammonium formate as B, while the negative mode mobile phase is made up of acetonitrile/water (60/40, v/v) with 10 mM ammonium acetate as A and 2-propanol/

acetonitrile (90/10, v/v) with 10 mM ammonium acetate as B. Both modes share the same gradient: 0–2 min from 15% to 30% B, 2–2.5 min from 30% to 48% B, 2.5–11 min from 48% to 82% B, 11–11.5 min from 82% to 99% B, 11.5–12 min maintain at 99% B, 12–12.1 min from 99% to 15% B, and 12.1–14.2 min re-equilibrate at 15% B. A ThermoFisher Q-Exactive HF with a HESI-II ion source (Thermo Scientific, Waltham, MA, USA) was used to collect spectra with a data-dependent MS/MS spectra acquisition method. The ion source conditions were set as follows: spray voltage, 3.6 kV; sheath gas flow rate, 60 arbitrary units; aux gas flow rate, 25 arbitrary units; sweep gas flow rate, 2 arbitrary units; capillary temp, 300 °C; S-lens RF level, 50; Aux gas heater temperature, 370 °C. The following acquisition parameters were used for MS1 analysis: resolution, 60,000, AGC target, 1e6; Maximum IT, 100 ms; scan range 150–1700$m/z$; spectrum data type, centroid. Data-dependent MS/MS parameters: resolution, 15,000; AGC target, 1e5; maximum IT, 50 ms; loop count, 4; TopN, 4; isolation window, 1.0$m/z$; fixed first mass, 70.0$m/z$; (N)CE/stepped nce, 20, 30, 40; spectrum data type, centroid; minimum AGC target, 8e3; intensity threshold, 1.6e5; exclude isotopes, on; dynamic exclusion, 3.0 s. To increase the total number of MS/MS spectra, five runs with iterative MS/MS exclusions were performed using the R package "IE-Omics"[18] for both positive and negative electrospray conditions.

**HILIC–MS/MS analysis**. For LC–MS/MS analysis of polar metabolites, the same ThermoFisher equipment was used as above. Three microliters of the resuspended HILIC solution was injected onto a Waters Acquity UPLC BEH Amide column (150 mm × 2.1 mm; 1.7 μm) coupled with an additional Waters Acquity VanGuard BEH Amide precolumn (5 mm × 2.1 mm; 1.7 μm). The oven temperature was maintained at 45 °C, and the flow rate was set at 0.4 mL/min. HILIC chromatographic separations were performed by the following parameters: solvent A consisted of water with 10 mM ammonium formate and 0.125% formic acid, solvent B was made from acetonitrile/water (95/5, v/v) with 10 mM ammonium formate and 0.125% formic acid. A gradient run was set up as 0–2 min at 100% B, 2–7.7 min from 100% to 70% B, 7.7–9.5 min from 70% to 40% B, 9.5–10.25 min from 40% to 30% B, 10.25–12.75 min from 30% to 100% B, and 12.75–17 min re-equilibrate at 100% B. Mass spectrometry parameters were identical as above, but the MS1 mass was limited to 60–900 m/z.

**GC-TOF MS analysis**. 0.5 μL sample was injected with 25 s splitless time on an Agilent 6890 GC (Agilent Technologies, Santa Clara, CA) using a Restek Rtx-5Sil MS column (30 m × 0.25 mm, 0.25 μm) with 10 m Guard column (10 m × 0.25 mm, 0.25 μm) and 1 mL/min Helium gas flow. The oven temperature was held 50 °C for 1 min, ramped up to 330 °C at 20 °C/min, and held for 5 min. Data were acquired at 70 eV electron ionization at 17 spectra/s from 85 to 500 Da at 1850 V detector voltage on a Leco Pegasus IV time-of-flight mass spectrometer (Leco Corporation, St. Joseph, MI). The transfer line temperature was held at 280 °C with an ion source temperature set at 250 °C. Standard metabolites mixtures and blank samples were injected at the beginning of the run and every 10 samples throughout the run for quality control. Raw data were preprocessed by ChromaTOF version 4.50 for baseline subtraction, deconvolution, and peak detection. Binbase was used for metabolite annotation and reporting[20].

**LC-MS data processing and statistics**. All the LC–MS raw data files were converted into ABF format using ABF converter (https://www.reifycs.com/AbfConverter/). MS-DIAL ver.4.00 software was used for deconvolution, peak picking, alignment, and compound identification[19]. The detailed parameter setting was as follows: MS1 tolerance, 0.005 Da; MS2 tolerance, 0.01 Da; minimum peak height, 20,000 amplitude; mass slice width, 0.1 Da; smoothing method, linear weighted moving average; smoothing level, 5 scans; minimum peak width, 10 scans. $[M + H]^+$, $[M + NH_4]^+$, $[M + Na]^+$, $[2M + H]^+$, $[2M + NH_4]^+$, $[2M + Na]^+$ were included in adduct ion setting for positive mode lipidomics and HILIC analysis, $[M-H]^-$, $[M + Cl]^-$, $[M + Hac-H]^-$ for negative mode lipidomics, and $[M–H]^-$, $[M + Cl]^-$, $[M + FA-H]^-$, $[2M-H]^-$ for negative mode HILIC analysis. Compounds were annotated by matching retention times, accurate precursor masses, and MS/MS spectra against libraries in MassBank of North America (https://mona.fiehnlab.ucdavis.edu/) and NIST17 (https://chemdata.nist.gov/). Retention time libraries were produced from authentic standards and extrapolated for lipids as published before[19]. The primary result data matrix was processed with MS-FLO software to identify ion adducts, duplicate peaks, and isotopic features[66]. Systematic error removal by random forest (SERRF software, https://slfan2013.github.io/SERRF-online/#)[28] was employed to correct for batch effects or instrument signal drifts. For metabolites that were detected by two or more platforms, values with the lowest relative standard deviation in quality control samples were kept. Metabolites that were present in at least 6 of the 8 samples in at least one of the 80 study groups (defined by age, sex, and brain region) were kept in the dataset, otherwise, metabolites were removed from the dataset. Missing data were replaced by 1/10th of the minimum value (default value 100).

Estimated concentrations were calculated based on a series of internal standards with known concentrations spiked during the sample preparation. The quantitative results of metabolites were obtained using the peak heights and the concentrations of the spiked internal standards and then normalized to sample fresh weight. For HILIC metabolites, the quantification was achieved by comparing to their corresponding isotope-labeled internal standards. For lipid quantification, the concentrations of all

lipid candidates in a lipid class were estimated by the corresponding internal standard of that class. As there was only one internal standard in each lipid class to quantify a large diversity of lipids, it should be noted that the accuracy of quantifications was inevitably affected by matrix effects and may not fully reflect the different MS responses of lipids with different fatty acyl chains. In addition, for those important neurochemicals whose internal standards were not available, the concentrations in the pooled QC samples were estimated according to the reported endogenous concentrations in the brain and then applied to all the brain samples. Quantification results and quantification methods are summarized in Supplementary Data 3.

Statistical analysis was performed by normalization to the median intensity of all identified compounds, log transformation, and Pareto scaling. PCA was used for multivariate statistics and visualization, specifically for outlier detection. From the total 640 total biological samples, three outlier samples were removed by outlier analysis in PCA plots, including one MD sample from a female early adult, one BG sample from a middle-aged female, and one OB from an old-aged male. Results from Kruskal–Wallis tests were followed by Dunn's multiple comparison confinement. Results from Mann–Whitney $U$ tests were corrected by the Benjamini–Hochberg procedure to control the false discovery rate. Spearman rank correlation analyses and fold change calculations were conducted using R.

**Reporting summary**. Further information on research design is available in the Nature Research Reporting Summary linked to this article.

## Data availability
The authors declare that data supporting the findings of this study are available within the paper and its Supplementary Information files. This data is available at the NIH Common Fund's National Metabolomics Data Repository (NMDR) website, the Metabolomics Workbench (https://www.metabolomicsworkbench.org) where it has been assigned Project ID PR001047. The data can be accessed directly via https://doi.org/10.21228/M8C68D.

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

## Acknowledgements

We thank Anand Parag Bakshi and Honglian Ye for sample preparation, Luis Valdiviez for GC-TOFMS data acquisition and data curation and Gert Wohlgemuth and Diego Pedrosa for help in deploying the web tool. This work was funded by NIH U2C ES030158 and U19 AG023122 (to O.F.), KOMP2-Phase 2 Production and Phenotyping by the DTCC Consortium (NIH UM1 OD023221, to K.C.K.L.), and a fellowship from the Postdoctoral International Exchange Program of China Postdoctoral Science Foundation to J.D. The Metabolomics Workbench project is supported by NIH grant U2C-DK119886.

## Author contributions

J.D., M.R.S. and O.F. designed the research. R.A., L.R.B. and K.C.K.L. performed the brain tissue collection. J.D. and Z.R. analyzed brain samples. J.D. performed data acquisition and data processing. T.S., J.F., Y.Z., and J.D. curated the MS spectra. J.J. developed the web tool and S.M. contributed to the web tool deployment. C.R.B. and X.L. performed the statistics. S.F. carried out data normalizations. J.D. and O.F. wrote the paper with contributions from all other authors.

## Competing interests

The authors declare no competing interests.
