## [Peer Review File · Nature Communications]

REVIEWER COMMENTS

Reviewer #1 (Remarks to the Author):

The authors presented an impressive dataset regarding metabolomic alterations in 10 brain anatomical regions at 3 time points (3,16, and 59 weeks) of normal mice. Based on multiple platforms, over 1,000 metabolites were identified that cover a wide spectrum of metabolites from nucleosides, lipids, glycolysis intermediates, to neurochemicals. A free-access web tool was also developed to allow interactive exploration of this dataset. Compared to existing animal studies on brain metabolome, this study provides new insights/ideas to brain structure and function with more brain regions, a wider coverage of metabolites, but shorter life span. However, the data analysis and interpretation is not clear or sufficiently strong to support some of the findings. More work should be conducted for a comprehensive and complete presentation of the study.

1. I would like to underscore the importance of data pretreatment especially for the case of employing three chromatography-based mass spectrometry platforms. The relevant descriptions in method section were too brief for me to justify the reliability of downstream results.

Why pareto but not frequently-used uv scaling was used?

Have missing values been imputed?

Have sparse variables been eliminated? And how?

How to select the variables if the same metabolite was identified by two or more platforms?

Have the confounders (e.g. gender, weight, cage index) been taken into consideration in the correlation analysis?

2. Two-way ANOVA may be a more appropriate method for this study compared with Kruskal-Wallis and Mann-Whitney U tests. Because there are at least two factors (weeks of age and brain region) affecting the brain metabolome simultaneously and there are interactions between them. All the relevant results and interpretations may change if a different or a suboptimal method is used.

3. It is fine to present results by chemical classes or pathways. I would appreciate if the authors also illustrate the alterations of metabolic clusters derived from some unsupervised methods (e.g. HCA) as metabolites from different chemical classes may shift in a synchronous manner during aging. There may be some new findings if cluster analysis is used.

4. As the authors claimed, there is minor gender difference (Fig S2a). What's the cause/meaning of the two clusters in the PCA scores plot? There are also two clusters in Fig 3a. Is this bias caused by batch effect or interesting findings with biological significance?

5. How to explain the apparent negative correlations between brainstem and cerebrum in Fig 3C?

6. Please extend Fig 1a by adding the scheme of bioinformatic analysis.

7. Study limitations. The authors have paid adequate attention to the limitations of their work. I would like to suggest expanding study limitations with the following points. (1) As an atlas, more time points cover the entire lifespan are needed. The oldest group in this study is only 59 weeks, roughly equivalent to a middle-aged person. In addition, blood and fecal metabolome may also provide auxiliary information on brain function. (2) Each group of rats were sacrificed at different time points in order to obtain brain, meaning data points of each age are from different animals. It's hard to correct or control this bias with the limited number of samples. (3) All the findings are based on rats and the underlying biochemical mechanisms remain poorly understood, we should be very careful when extrapolating these preliminary findings to humans.

8. The authors should validate some of the highlighted findings (fig 4d, fig 5b and 5c) by targeted quantitative data.

Reviewer #2 (Remarks to the Author):

In the presented manuscript the authors provide an extensive report of annotated complex lipidome and polar metabolome across ten brain regions and explore the variation patterns of identified (polar

and lipid) metabolites as a function of brain region, mouse age and sex. For comprehensive metabolome profiling, the authors have combined two technological platforms (LC-MS and GC-MS) and three different analytical assays: one using reversed phase chromatography for complex lipid analysis (in ESI+ and ESI- mode) and two, one using HILIC (also in ESI+ and ESI- mode) and one using GC-MS for polar metabolite analysis. Combining these three assays authors have successfully identified around 1200 complex lipids and 500 polar metabolites. The results of these analyses can be interactively explored using the developed web application – as a dynamic analysis report, one can get an overview of metabolic profiles per metabolite, per region and mouse age. Although this manuscript provides a comprehensive atlas of metabolome changes across the heterogenous brain tissue, this study remains fairly descriptive with several efforts to explain the observed changes (across brain metabolome) using the previously created gene and protein atlases. In addition to its descriptive character, several other issues remain.

One major concern is related to the explored time points along the mouse lifespan. It would have been essential to have at least one additional time point for an aged mouse brain, at 24 months - as suggested by authors themselves in the conclusive part of the manuscript. This would be highly relevant with respect to metabolic alterations occurring with age – as a major risk factor for the acquired neurodegenerative diseases.

The authors should specify what the reported proportions of different metabolite classes refer to – a priori they discuss the proportions with respect to metabolite diversity (i.e. numbers of annotated metabolites) and not the metabolite quantities or concentrations. From the acquired data (peak areas in ion counts), the authors cannot conclude about the metabolite proportions in the overall metabolome. Brain tissue is for sure rich in lipids but lipids also ionize very well resulting in highly abundant signals – which do not necessarily translate that they are present in highest concentrations (but rather their high ionization efficiencies). This should also be specified in the Figure 1. – that the reported chemical composition reflects the metabolite diversity – in terms of annotated metabolite species (but not their quantities). In general, the quantitative data are lacking. For a metabolome atlas of aging brain, it would have been important to report the metabolite concentrations (normalized to proteins). While this would be difficult to achieve for the entire reported metabolite set, it could be done at least for specific metabolite classes.

The paragraph on data quality assessment (including metabolite CVs and PCA) should be removed to the materials and methods section. The applied strategy is commonly used in untargeted metabolomic experiments, the signal intensity drift correction using QC samples is mandatory prior to the exploration of biologically relevant changes.

Several minor concerns:

In general, the authors should make an effort to homogenize the terms they are using – brain regions or sections? Why the term sections is used in the abstract? In the same line, they should use the same logic to label the analytical assays, for complex lipid analysis – RPLC ESI + or ESI-, for polar metabolite analysis HILIC ESI+ or ESI-. For some reason the column name “CSH” is used for complex lipid analysis but not for polar metabolite analysis.

The authors should not talk about metabolite expression, this term is used for genes and proteins. Metabolites are present in lower or higher levels, metabolite levels vary but not their “expression”.

In the supplementary table 2, the authors report peak heights. Were the peak heights used for statistical analysis? Or rather peak areas?

How was confirmed the identity of polar metabolites for which the MS/MS data is lacking?

Reviewer #3 (Remarks to the Author):

This is an outstanding resource for the fields of neuroscience and aging. I have no concerns with the data or writing of the manuscript. However, in my opinion it would have been valuable to include a group of very old mice (20-24 months of age) in the study.

Reviewer(s)' Comments to Author:

Reviewer #1 (Remarks to the Author):

The authors presented an impressive dataset regarding metabolomic alterations in 10 brain anatomical regions at 3 time points (3,16, and 59 weeks) of normal mice. Based on multiple platforms, over 1,000 metabolites were identified that cover a wide spectrum of metabolites from nucleosides, lipids, glycolysis intermediates, to neurochemicals. A freeaccess web tool was also developed to allow interactive exploration of this dataset. Compared to existing animal studies on brain metabolome, this study provides new insights/ideas to brain structure and function with more brain regions, a wider coverage of metabolites, but shorter life span.

Reviewer 1: However, the data analysis and interpretation is not clear or sufficiently strong to support some of the findings. More work should be conducted for a comprehensive and complete presentation of the study.

REPLY BY THE AUTHORS: We have added old age mouse data now and we have extended our data analysis and interpretation.

Reviewer 01, Q01 "Methods".

Q01a: I would like to underscore the importance of data pretreatment especially for the case of employing three chromatography-based mass spectrometry platforms. The relevant descriptions in method section were too brief for me to justify the reliability of downstream results.

REPLY BY THE AUTHORS: More detailed information was added in the LC-MS Data processing and statistics Section of the revised manuscript.

Q01b: Why pareto but not frequently-used uv scaling was used?

REPLY BY THE AUTHORS: Pareto scaling uses the square root of the standard deviation as the scaling factor instead of the standard deviation itself in comparison to UV scaling. It is an intermediate between UV and no scaling, as it up weights medium features without inflating base line noise. Data by Pareto scaling commonly stays closer to the original measurement than UV scaling. Therefore, Pareto scaling is recommended in metabolomics and employed in our study.

Q01c: Have missing values been imputed?

REPLY BY THE AUTHORS: Yes, and we have added that information to the method section. GC-TOF MS did not have missing values. In GC-TOF MS data processing, compounds that were positively detected in some chromatograms, but not in others, were automatically replaced from raw data (of the target mass at that target retention time) during the data processing procedure. Hence, formerly missing values were automatically replaced by local noise. For other assays, missing values had to be replaced because for high-resolution MS data, accurate masses may have zero values in raw data, i.e. noise values are difficult to estimate. Instead, we replaced all missing values by very low "100" values in the dataset, representing less than 1% of genuine values.

Q01d: Have sparse variables been eliminated? And how?

REPLY BY THE AUTHORS: Yes, and we have added that information to the method section.

Metabolites that were present in at least 6 of the 8 samples in at least one of the 80 study groups (defined by age, sex and brain region) were kept in the dataset, otherwise metabolites were removed from the dataset.

Q01e: How to select the variables if the same metabolite was identified by two or more platforms?

REPLY BY THE AUTHORS: We have added that information to the method section. For metabolites that were detected by two or more platforms, values with the lowest relative standard deviation in quality control samples were kept.

Q01f: Have the confounders (e.g. gender, weight, cage index) been taken into consideration in the correlation analysis?

REPLY BY THE AUTHORS: We do not think confounders are the correct term here. Confounders are important to consider for classic univariate statistics to investigate averages such as Kruskal-Wallis. Correlations do not consider averages, but instead, correlations depend on variance between animals. Yet, of course the eight mice per age and sex group did not show noticeable differences in weight. Sex was a minor component of overall variance; that is why we show unsupervised PCA plots instead of supervised PLS-DA. The UC Davis mouse biology program is one of the largest service facilities in the United States and take utmost care to avoid any bias by cages, feed, or other possible confounders in animal housing.

Q01g: Two-way ANOVA may be a more appropriate method for this study compared with Kruskal-Wallis and Mann-Whitney U tests. Because there are at least two factors (weeks of age and brain region) affecting the brain metabolome simultaneously and there are interactions between them. All the relevant results and interpretations may change if a different or a suboptimal method is used.

REPLY BY THE AUTHORS: We do not agree with this statement of testing interaction effects. First of all, we are not interested in the general question if metabolites are different between any combination of two factors (age and brain region). Similarly, we are not interested in testing if a metabolite in “pons at adolescence” is different to “cerebellum at old age”. That again is not interesting biologically. We are mostly interested in pairwise comparisons of adjacent age groups (e.g. middle age and old age), or in pairwise comparisons of brain regions within one age group. These statistical questions are biologically more relevant.

Secondly, two-way ANOVAs would appropriate if the data met the assumptions of normality, homogeneity of variances and independence. However, metabolomics data are not normal distributed and have unequal variances. Non-parametric tests such as the Kruskal-Wallis and Mann-Whitney U test are the appropriate tests in metabolomics. Specifically, in our study nearly 30% annotations were not normal distributed, and non-parametric tests were employed.

Reviewer 01, Q02. It is fine to present results by chemical classes or pathways. I would appreciate if the authors also illustrate the alterations of metabolic clusters derived from some unsupervised methods (e.g. HCA) as metabolites from different chemical classes may shift in a synchronous manner during aging. There may be some new findings if cluster analysis is used.

REPLY BY THE AUTHORS: We exclusively used unsupervised methods throughout our analyses. Unsupervised hierarchical clustering/heatmaps are given in Supplementary Figure 4 to illustrate the shift of metabolite patterns during aging.

Reviewer 01, Q03. As the authors claimed, there is minor gender difference (Fig S2a). What's the cause/meaning of the two clusters in the PCA scores plot? There are also two clusters in Fig 3a. Is this bias caused by batch effect or interesting findings with biological significance?

REPLY BY THE AUTHORS: The two clusters in the PCA plots of Figure S2A and 3A are adolescent brain samples (the top cluster), versus early adults and middle age samples (the bottom cluster). They were indicated in Figure 2A. For clarity, the PCA plots in the previous Figure S2A, 2A and 3A were the same plots but colored by different groups to illustrate the impact of sexes, ages, and brain regions. With the new dataset containing 92-week-old mouse brains, there are now four clusters to discriminate different ages in PC1, which were labeled as adolescent, early adults, middle age and old age, respectively in Figure 3B of the revised manuscript.

Reviewer 01, Q04. How to explain the apparent negative correlations between brainstem and cerebrum in Fig 3C?

REPLY BY THE AUTHORS: Negative correlations between brainstem and cerebrum indicate there is a large difference in metabolic regulations between the two brain divisions. Hence, the topographical and functional difference in these large brain regions is also supported by regionally specific metabolism, as shown in Figure 2D as well. It is one of the main discoveries of such a Metabolome Atlas. Negative metabolomic correlations between different regions could mirror the differences in functions at the physiological level as well. The cerebrum is mostly in charge of higher cognitive functions, while the brainstem regulates visceral activities. Therefore, different sets of metabolites are the essential elements to perform the corresponding functions. Similar results were also reported by a previous study of mouse brain gene expression atlas (Nature 2007, 445 (7124), 168-176.). We added these explanations as discussion in the revised manuscript to make that important feature more clear.

Reviewer 01, Q05. Please extend Fig 1a by adding the scheme of bioinformatic analysis.

REPLY BY THE AUTHORS: Figure 1A was extended in the revised manuscript.

Reviewer 01, Q06. Study limitations. The authors have paid adequate attention to the limitations of their work. I would like to suggest expanding study limitations with the following points. (1) As an atlas, more time points cover the entire lifespan are needed. The oldest group in this study is only 59 weeks, roughly equivalent to a middle-aged person. In addition, blood and fecal metabolome may also provide auxiliary information on brain function. (2) Each group of rats were sacrificed at different time points in order to obtain brain, meaning data points of each age are from different animals. It's hard to correct or control this bias with the limited number of samples. (3) All the findings are based on rats and the underlying biochemical mechanisms remain poorly understood, we should be very careful when extrapolating these preliminary findings to humans.

REPLY BY THE AUTHORS: Revisions were made according to these comments. (1) Most importantly, we have now added old age mice. Eight mice per age and sex group is very commonly used in mouse model research. We agree that the brain atlas can only be a start. We are discussing this topic with the NIH, and we are aware of other efforts in Europe (not on aging but on dietary impact on mice). (2) As we have only found limited differences in brain metabolism between the sexes, they can be summarized for most age-dependent comparisons, clearly increasing statistical power. Mice were held in adjacent cages in the same room, having the same

caretakers, the same animal feed, the same environment and the same access to physical activity. Hence, most typical biases in mouse biology have been taken care of, but of course, additional unknown biases may always be a concern and can never be completely ruled out. (3) Mice are not humans. We think it would be a valid hypothesis that similar brain aging phenotypes could be observable in rats, but we have not tested that hypothesis. We certainly agree that rodents are not primates, either, and understand the caveats when extrapolating findings from one species to the next. Our next “atlas” will be a comparison of plasma metabolomics across 17 vertebrates at early adult age, including different primates and humans, but more careful comparisons will be deliberately and carefully discussed with the scientific community at large (and funded by a dedicated funding mechanism).

Reviewer 01, Q07. The authors should validate some of the highlighted findings (fig 4d, fig 5b and 5c) by targeted quantitative data.

REPLY BY THE AUTHORS: We have performed quantitative analysis for 853 metabolites in the aging mouse brain metabolome atlas. Results are now summarized in Supplementary Table 3. The endogenous concentrations of dopamine, adenosine, guanine, HexCers, sHexCers and SMs in the above-mentioned Figures are all included in the supplement data. Specifically, the quantification of dopamine, adenosine and guanine were carried out by adding their corresponding isotope labeled internal standard in each sample. Because there are many chemical species in each lipid class, it is unrealistic to obtain an internal standard for each lipid. Instead, the most commonly used approach for quantitation in lipidomics is to use a single internal standard as representative for classes of lipids. To this end, we employed glucosyl ceramide(d18:1/17:0), mono-sulfo galactosyl ceramide(d18:1/17:0) and SM (18:1/17:1) as the internal standard of HexCer, sHexCer and SM species for quantification. These odd-chain surrogate internal standards are not endogenously present in the brain. In addition, we validated the results by comparing the concentrations of some representative metabolites to previously published papers. Results were found to be consistent, suggesting a good accuracy of our findings.

Reviewer #2 (Remarks to the Author):

In the presented manuscript the authors provide an extensive report of annotated complex lipidome and polar metabolome across ten brain regions and explore the variation patterns of identified (polar and lipid) metabolites as a function of brain region, mouse age and sex. For comprehensive metabolome profiling, the authors have combined two technological platforms (LC-MS and GC-MS) and three different analytical assays: one using reversed phase chromatography for complex lipid analysis (in ESI+ and ESI- mode) and two, one using HILIC (also in ESI+ and ESI- mode) and one using GC-MS for polar metabolite analysis. Combining these three assays authors have successfully identified around 1200 complex lipids and 500 polar metabolites. The results of these analyses can be interactively explored using the developed web application – as a dynamic analysis report, one can get an overview of metabolic profiles per metabolite, per region and mouse age. Although this manuscript provides a comprehensive atlas of metabolome changes across the heterogenous brain tissue, this study remains fairly descriptive with several efforts to explain the observed changes (across brain metabolome) using the previously created gene and protein atlases. In addition to its descriptive character, several other issues remain.

REPLY BY THE AUTHORS: Thank you for the comments.

Reviewer 02, Q01: One major concern is related to the explored time points along the mouse lifespan. It would have been essential to have at least one additional time point for an aged mouse brain, at 24 months - as suggested by authors themselves in the conclusive part of the manuscript. This would be highly relevant with respect to metabolic alterations occurring with age – as a major risk factor for the acquired neurodegenerative diseases.

REPLY BY THE AUTHORS: We agree to this point and have added 92-week-old (21-month-old) mice to the study, which is equivalent to 71-year-old humans. After this time point, mice start to die at accelerated age, indicating a phase of rapid deterioration in general health status similar as observed in humans. As we think such phase deserves specific investigations, including for the onset of acquired neurodegenerative diseases, such age groups were not included in this generic atlas study. We dissected the mouse brains, employed the exactly same method to acquire metabolomics data of the old mouse brains, and combined the two batches of data. Importantly, quality control figures showed no batch effect in our study (Figure 2), with the quality control pool samples centered nicely with very limited variance in the middle of the PCA plots. Based on the new dataset of aging mouse brain, all Figures and Tables were updated. Findings and discussions were extended to the old-aged mouse brains in the revised manuscript.

Reviewer 02, Q02: The authors should specify what the reported proportions of different metabolite classes refer to – a priori they discuss the proportions with respect to metabolite diversity (i.e. numbers of annotated metabolites) and not the metabolite quantities or concentrations. From the acquired data (peak areas in ion counts), the authors cannot conclude about the metabolite proportions in the overall metabolome. Brain tissue is for sure rich in lipids but lipids also ionize very well resulting in highly abundant signals – which do not necessarily translate that they are present in highest concentrations (but rather their high ionization efficiencies). This should also be specified in the Figure 1. – that the reported chemical composition reflects the metabolite diversity – in terms of annotated metabolite species (but not their quantities).

REPLY BY THE AUTHORS: In the revised manuscript, we extended the Figure 1 caption to explain that the reported chemical composition reflects the metabolite diversity. We want to emphasize our study design for which we have used three different assays: lipidomics, biogenic amines and primary metabolites. For the latter two assays, lipids were removed prior to analyses (through solvent extraction / fractionation). Hence, potential problems in ion suppression or the ability of compounds to be ionized were limited due to the use of HILIC-MS/MS (including positive and negative electrospray) and the complementary use of GC-electron (impact) ionization-MS, a completely different ionization mechanism. Yet, of course the authors agree that every analytical technique has its blind spots and its biases – there is no 'unbiased' analysis, as papers in the early years of metabolomics might have promised. However, due to the combination of fractionation and different separation and ionization techniques, such biases are more limited than if only a single technique would have been used. These arguments are added into Method section.

Reviewer 02, Q03: In general, the quantitative data are lacking. For a metabolome atlas of aging brain, it would have been important to report the metabolite concentrations (normalized to proteins). While this would be difficult to achieve for the entire reported metabolite set, it could be done at

least for specific metabolite classes.

REPLY BY THE AUTHORS: We have performed quantitative analysis for 853 metabolites in the aging mouse brain metabolome atlas. Results are now summarized in Supplementary Table 3. The endogenous concentrations of dopamine, adenosine, guanine, HexCers, sHexCers and SMs in the above-mentioned Figures are all included in the supplement data. Specifically, the quantification of dopamine, adenosine and guanine were carried out by adding their corresponding isotope labeled internal standard in each sample. Because there are many chemical species in each lipid class, it is unrealistic to obtain an internal standard for each lipid. Instead, the most commonly used approach for quantitation in lipidomics is to use a single internal standard as representative for classes of lipids. To this end, we employed glucosyl ceramide(d18:1/17:0), mono-sulfo galactosyl ceramide(d18:1/17:0) and SM (18:1/17:1) as the internal standard of HexCer, sHexCer and SM species for quantification. These odd-chain surrogate internal standards are not endogenously present in the brain. In addition, we validated the results by comparing the concentrations of some representative metabolites to previously published papers. Results were found to be consistent, suggesting a good accuracy of our findings.

Reviewer 02, Q04: The paragraph on data quality assessment (including metabolite CVs and PCA) should be removed to the materials and methods section. The applied strategy is commonly used in untargeted metabolomic experiments, the signal intensity drift correction using QC samples is mandatory prior to the exploration of biologically relevant changes.

REPLY BY THE AUTHORS: While we would generally agree to this suggestion, in this specific case we would like to urge leaving the QC Figure in the main manuscript, for the following reasons: (a) as an atlas is likely to serve as data source for other studies to be compared to, the quality of the data matters more with respect to build trust and to be displayed than for studies that only focus on a specific biological phenomenon. (b) More importantly, perhaps, is the fact that we have now added a completely new group of mice at 92-week old, with newly acquired metabolomic data. In some corners of the scientific and biomedical community, metabolomics has a reputation of not being repeatable. Figure 2, and specifically the QC samples, shows how excellently the two independently acquired data sets were repeatable. Fortunately enough, we had saved enough material from the first batch of QC samples to be run along with the second installment of data acquisitions. Otherwise, we would not have been able to showcase this success. Indeed, we are pretty proud of this success, given that the first data set was entirely acquired by first author Dr. Jun Ding who subsequently relocated to Wuhan, China, whereas the second data set was entirely acquired by our UC Davis technical staff directed by last author Dr. Fiehn under constant, but remote, advice by Dr. Ding from China using emails and zoom. Seriously, that is not something you would find in every lab. We don't think these considerations belong to either the manuscript or the supplements, but in fact, we actually thought about spinning off an entirely different manuscript only on this technical success. Now, with every success come some caveats: in the second dataset, we failed to detect about 162 metabolites, so the total number of identified metabolites decreased from 1709 to 1547 compounds. This was expected for such an untargeted metabolomics approach on three assay platforms. Still, this number of identified metabolites is much larger than for any other published brain metabolomics paper.

Reviewer 02, Q05: Several minor concerns:

Q05a: In general, the authors should make an effort to homogenize the terms they are using – brain regions or sections? Why the term sections is used in the abstract? In the same line, they should use the same logic to label the analytical assays, for complex lipid analysis –RPLC ESI+ or ESI-, for polar metabolite analysis HILIC ESI+ or ESI-. For some reason the column name “CSH” is used for complex lipid analysis but not for polar metabolite analysis.

REPLY BY THE AUTHORS: We agree. We now use the wording ‘brain regions’ throughout the manuscript. Complex lipid analysis is now detailed as reversed phase liquid chromatography, in line with other methods. Yet, both lipidomics and polar metabolites use a wide array of different columns and buffers. Previously, we published comparisons between different lipidomics techniques including the CSH column (charged surface hybrid), and different LC buffers. Curiously, such comparisons have never been published for HILIC methods, but CSH certainly is not useful for HILIC for which we here used the Waters Acquity UPLC BEH Amide column (‘bridged ethylene hybrid column’, i.e. an entirely different column chemistry than the CSH method.)

Q05b: The authors should not talk about metabolite expression, this term is used for genes and proteins. Metabolites are present in lower or higher levels, metabolite levels vary but not their “expression”.

REPLY BY THE AUTHORS: We agree. We now use ‘metabolite profile’ or ‘metabolite levels’ to replace ‘metabolite expression’ in the revised manuscript.

Q05c: In the supplementary table 2, the authors report peak heights. Were the peak heights used for statistical analysis? Or rather peak areas?

REPLY BY THE AUTHORS: Only the peak heights were used for statistical analysis. Over much of the span of calibration curves, peak heights and peak areas are giving exactly the same results. However, for very abundant peaks, peak areas give better accuracy and precision, while for very low abundant peaks, peak heights give better accuracy and precision. The reason is because for peak area integrations of low abundant peaks in experimental analyses, the slope of the baseline, the start/end of a peak and possible co-elutions yield large technical errors. For peak height determinations, these three problems matter much less. Because there are many more low abundant signals than super-abundant signals in metabolomics, we used peak heights rather than peak areas for quantitative determinations.

Q05d: How was confirmed the identity of polar metabolites for which the MS/MS data is lacking?

REPLY BY THE AUTHORS: We have published the use of accurate retention time (RT) matching on top of accurate mass MS and MS/MS matching before, for example, for prediction of retention times. We have also validated the Fiehnlab HILIC library comprised of accurate mass, retention time and MS/MS spectra for over 1,500 polar metabolites to be repeatable and reproducible at less than 6 seconds absolute RT difference after column or buffer changes, and less than 2 seconds relative RT differences after RT correction using our set of internal standards (That part is yet unpublished...we will submit that manuscript separately). Hence, RT plus accurate mass matches are two reliable, orthogonal measures of compound identity for our specific method and our specific library of authentic chemical standards. We agree that adding MS/MS is even better for confidence, and we have clarified the level of confidence in the column of “MSI level” of Supplementary Table 2 for each annotation. However, data dependent MS/MS sometimes misses to fragment ions, that

is the curse of cycle time in DDA mass spectrometry (for comparison, DIA, sometimes called IDA, gives too convoluted MSMS spectra for reliable identification of small molecules). The Fiehn HILIC library was uploaded to Massbank of North America (MoNA) for public use, so we do not call this an "in-house" library.

Reviewer #3 (Remarks to the Author):

Reviewer 03, Q01: This is an outstanding resource for the fields of neuroscience and aging. I have no concerns with the data or writing of the manuscript. However, in my opinion it would have been valuable to include a group of very old mice (20-24 months of age) in the study.

REPLY BY THE AUTHORS: We have now included 92-week-old (21-month-old) mice in the study, which is equivalent to 71-year-old humans. We updated the dataset and all the figures. The new findings on the old-aged mouse brains were also added in the revised manuscript.

REVIEWERS' COMMENTS

Reviewer #1 (Remarks to the Author):

This is an impressive resource for the fields of metabolomics and neuroscience. The authors have addressed all of my concerns and I have no further comments/suggestions on the data or writing of the manuscript.

Reviewer #2 (Remarks to the Author):

The authors have considered all the major concerns and adequately updated their manuscript. The integration of an additional time point (i.e. analysis of aged mouse brain) was performed without an issue with a batch effect, which is quite remarkable. The authors' sincerity about a portion of metabolites which were not detected (and therefore discarded) in this additional batch of samples is greatly appreciated, and effectively a panel of 1547 identified polar and lipid metabolites is fairly comprehensive.

One major concern remains, with utilization of internal standards for metabolite quantification. The authors claim that "Because there are many chemical species in each lipid class, it is unrealistic to obtain an internal standard for each lipid. Instead, the most commonly used approach for quantitation in lipidomics is to use a single internal standard as representative for classes of lipids." However, the authors have omitted the fact that this approach is used only for DIA and/or HILIC-facilitated analysis of lipids, because these techniques allow for the co-elution of internal standards and endogenous lipids and therefore, accurate correction of matrix effects. The accurate quantification of lipid species using RPLC cannot be achieved with a single internal standard representative of each class of lipids. On top of this, the odd-chain surrogate internal standards have been abandoned in the lipidomics field due to many recent studies revealing their endogenous origin, mainly as products of microbial metabolism (hosted by every model organism). This is why recently released mixtures of internal standards are isotopically labeled, such as the one recently developed by Avanti. Even more, this mixture contains multiple IS for each lipid class, with varied fatty acid composition to account for the degree of unsaturation and fatty acid chain length. With respect to these issues, the authors should replace the term "absolute quantification" by the term "estimated quantification".

It is fairly important that a group who is at the origin of great analytical methods for qualitative analysis of complex lipids promotes the example of correct quantification protocol among young researchers. The authors should discuss the bias introduced (due to the impossibility to cover the entire panel of highly diverse lipid species with corresponding internal standards) and/or provide only the semi-quantitative data for lipids (rather than over- and/or under-estimated concentrations).

Line 523: "Estimates of absolute quantifications" is contradictory, please replace with "Estimated concentrations were calculated..."

Reviewer(s)' Comments to Author:

Reviewer #1 (Remarks to the Author):

This is an impressive resource for the fields of metabolomics and neuroscience. The authors have addressed all of my concerns and I have no further comments/suggestions on the data or writing of the manuscript

REPLY BY THE AUTHORS: We thank the reviewer for the insightful feedback during the review process and the positive comments.

Reviewer #2 (Remarks to the Author):

The authors have considered all the major concerns and adequately updated their manuscript. The integration of an additional time point (i.e. analysis of aged mouse brain) was performed without an issue with a batch effect, which is quite remarkable. The authors' sincerity about a portion of metabolites which were not detected (and therefore discarded) in this additional batch of samples is greatly appreciated, and effectively a panel of 1547 identified polar and lipid metabolites is fairly comprehensive.

REPLY BY THE AUTHORS: Thank you for constructive remarks and suggestions during the review process.

Q01: One major concern remains, with utilization of internal standards for metabolite quantification. The authors claim that "Because there are many chemical species in each lipid class, it is unrealistic to obtain an internal standard for each lipid. Instead, the most commonly used approach for quantitation in lipidomics is to use a single internal standard as representative for classes of lipids." However, the authors have omitted the fact that this approach is used only for DIA and/or HILIC-facilitated analysis of lipids, because these techniques allow for the co-elution of internal standards and endogenous lipids and therefore, accurate correction of matrix effects. The accurate quantification of lipid species using RPLC cannot be achieved with a single internal standard representative of each class of lipids. On top of this, the odd-chain surrogate internal standards have been abandoned in the lipidomics field due to many recent studies revealing their endogenous origin, mainly as products of microbial metabolism (hosted by every model organism). This is why recently released mixtures of internal standards are isotopically labeled, such as the one recently developed by Avanti. Even more, this mixture contains multiple IS for each lipid class, with varied fatty acid composition to account for the degree of unsaturation and fatty acid chain length. With respect to these issues, the authors should replace the term "absolute quantification" by the term "estimated quantification".

REPLY BY THE AUTHORS: We agree that there are compromises of quantification accuracy by using a single internal standard as representative to quantify lipids, and the odd-chain species might not be an ideal internal standard due to the possibility of endogenous existence in the samples. Therefore, it is more appropriate to consider the quantitative results as "semi-quantification" in the study. We now replaced the term "absolute quantification" and "absolute concentrations" by "estimated quantification" and "estimated concentrations" in the revised manuscript.

Q02: It is fairly important that a group who is at the origin of great analytical methods for qualitative analysis of complex lipids promotes the example of correct quantification protocol among young researchers. The authors should discuss the bias introduced (due to the impossibility to cover the entire panel of highly diverse lipid species with corresponding internal standards) and/or provide only the semi-quantitative data for lipids (rather than over- and/or under-estimated concentrations).

REPLY BY THE AUTHORS: We have added more explanations in the Method section to discuss the bias introduced in the lipid quantification now. Please see in Line 531-534 in the revised manuscript.

Q03: Line 523: "Estimates of absolute quantifications" is contradictory, please replace with "Estimated concentrations were calculated..."

REPLY BY THE AUTHORS: The corresponding sentence was revised to "Estimated concentrations were calculated...".